# Thermodynamics of freezing and melting

Ulf R. Pedersen[1], Lorenzo Costigliola[1], Nicholas P. Bailey[1], Thomas B. Schrøder[1] & Jeppe C. Dyre[1]

Although the freezing of liquids and melting of crystals are fundamental for many areas of the sciences, even simple properties like the temperature–pressure relation along the melting line cannot be predicted today. Here we present a theory in which properties of the coexisting crystal and liquid phases at a single thermodynamic state point provide the basis for calculating the pressure, density and entropy of fusion as functions of temperature along the melting line, as well as the variation along this line of the reduced crystalline vibrational mean-square displacement (the Lindemann ratio), and the liquid's diffusion constant and viscosity. The framework developed, which applies for the sizable class of systems characterized by hidden scale invariance, is validated by computer simulations of the standard 12-6 Lennard-Jones system.

[1] Glass and Time, IMFUFA, Department of Science and Environment, Roskilde University, P. O. Box 260, Roskilde DK-4000, Denmark. Correspondence and requests for materials should be addressed to U.R.P. (email:ulf@urp.dk).

Melting is the prototypical first-order phase transition[1–3]. Its qualitative description has been textbook knowledge for a century, but it has proven difficult to give quantitatively accurate predictions. This is the case not only for the kinetics of freezing and melting, which are exciting and highly active areas of research[4–8]; there is not even a theory for calculating, for example, the entropy of fusion as a function of temperature along the melting line in the thermodynamic phase diagram.

The everyday observation that matter sticks together but is at the same time almost impossible to compress[9] is modelled, for example, in the system proposed by Lennard-Jones (LJ) in 1924 (ref. 10). Here, particles interact via a pair potential that as a function of distance $r$ is a difference of two inverse power-law terms: $v_{LJ}(r) = 4\varepsilon((r/\sigma)^{-12} - (r/\sigma)^{-6})$. The first term reflects the fact that the repulsive 'Pauli' forces are harsh and short-ranged, the negative term models the softer, longer ranged attractive van der Waals forces. The 1970s led to the development of highly successful thermodynamic perturbation and integral-equation theories for simple liquids[11–16]. Their main ingredient is the assumption that the structure of a dense, monatomic fluid closely resembles that of a collection of hard spheres[14,16–18]. Confirming this, the structure of melts of, for example, metallic elements near freezing is close to that of the hard-sphere system[15,16,18,19]. The term 'structure' generally refers to the entire collection of spatial equal-time density correlation functions, but our focus below is on the pair correlation function (in the form of its Fourier transform, the structure factor) as the most important structural characteristic.

Since the hard-sphere system has only a single nontrivial thermodynamic state parameter, the packing fraction, the phase diagram is basically one-dimensional, which implies that the system has a unique freezing/melting transition. On the basis of this, for simple systems one expects invariance along the freezing and melting lines of structure and dynamics in proper units, as well as of thermodynamic variables like the relative density change upon melting and the melting entropy[20]. Empirical freezing and melting rules, which follow from the hard-sphere melting picture and are fairly well obeyed for most simple systems, include the fact that the ratio between the crystalline root-mean-square atomic displacement and the nearest-neighbor distance—known as the Lindemann ratio—is constant and about 0.1 along the melting line; this is the famous Lindemann melting criterion from 1910 (refs 20–25). In the hard-sphere model the Lindemann ratio is universal at melting because, as mentioned, there is just a single melting point. Thus, for systems well described by the hard-sphere model the Lindemann ratio is predicted to be invariant along the melting line. Other empirical rules, which are predicted by the hard-sphere picture and reasonably well obeyed by many systems, include the facts that in properly reduced units the liquid's self-diffusion constant and viscosity are invariant along the freezing line[26,27], the Hansen–Verlet rule[17,28] that the amplitude of the first peak of the liquid static structure factor is about 2.85 at freezing, or Richard's melting rule[3] that the entropy of fusion $\Delta S_{fus}$ is about $1.1 k_B$ (which in a more modern and accurate version is the fact that the constant-volume entropy difference across the density–temperature coexistence region is close to $0.8 k_B$ (refs 23,29)).

The below study shows how the thermodynamics of freezing and melting for a large class of systems may be predicted to a good approximation from computer simulations carried out at a single coexistence state point. In particular, the theory developed quantifies the deviations from the above mentioned hard-sphere predicted melting-line invariants[16,22,30–32]. The theory is validated by computer simulations of the standard 12-6 LJ system.

## Results

**General theory.** It is well-known that adding a mean-field attractive term to the hard-sphere model broadens the coexistence region, which on the other hand, narrows if the repulsive part is softened[13,16,33–36]. Such terms are therefore expected to modify the hard-sphere predicted invariances along the freezing and melting lines. As an illustration, Fig. 1a shows that in reduced units there is approximate identity of structure along the LJ freezing line, but the structure is not entirely invariant as seen in the inset where the dashed line marks the predicted maximum based on simulations at $T_0 = 2.0\varepsilon/k_B$, if the structure were invariant.

In order to develop a quantitative theory of freezing and melting, we take as starting point the 'hidden scale invariance' property of systems[38] characterized by a potential-energy function $U(\mathbf{R})$, where $\mathbf{R} = (\mathbf{r}_1, \mathbf{r}_2, \ldots, \mathbf{r}_N)$ is the collective coordinate of the system's $N$ particles, which to a good approximation obeys the scaling condition.[39]

$$U(\mathbf{R}_a) < U(\mathbf{R}_b) \Rightarrow U(\lambda\mathbf{R}_a) < U(\lambda\mathbf{R}_b). \quad (1)$$

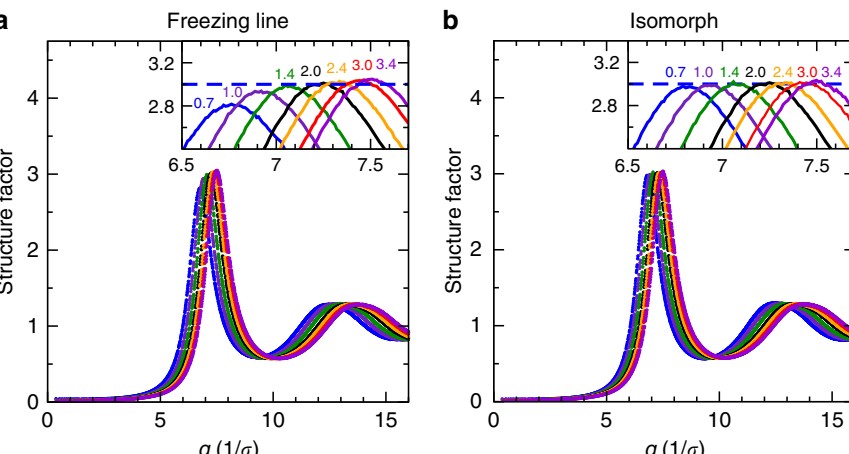

**Figure 1 | Structure of the LJ liquid.** (**a**) Liquid structure factor along the freezing line[37] showing results from $T = 0.7\varepsilon/k_B$, which is close to the triple point, to $T = 3.4\varepsilon/k_B$. The hard-sphere model predicts that the height of the first peak is invariant along the freezing line as indicated by the blue dashed line in the inset. Small, but systematic deviations are observed. (**b**) Liquid structure factor along the isomorph crossing the freezing line at temperature $T_0 = 2.0\varepsilon/k_B$ (henceforth used as the liquid reference isomorph), demonstrating structural invariance to a much higher degree. This is the basis for the theory proposed in the present paper.

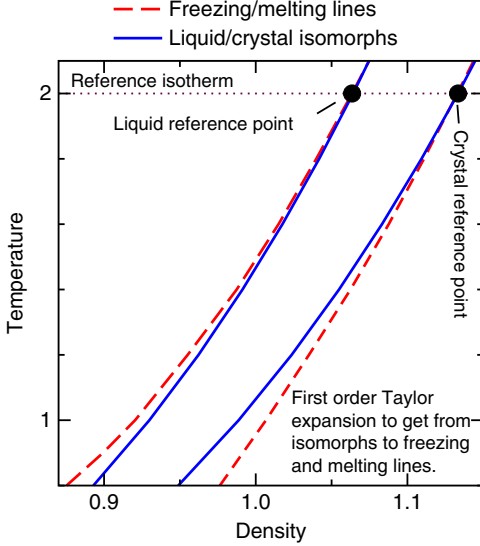

**Figure 2 | Illustration of the main idea of the theory.** The freezing and melting lines are both close to isomorphs along which basically everything is known because the reduced-unit structure and dynamics are invariant to a very good approximation. Properties along the freezing and melting lines are estimated via first-order Taylor expansions by moving from an isomorph to the freezing or melting line; the two reference isomorphs (a liquid and a solid one) are determined from computer simulations at $T_0 = 2.0\varepsilon/k_B$. Details are given in the 'Methods' section.

Here, $\lambda$ is a scaling factor and it is understood that the sample container undergoes the same scaling as the configuration; thus $\lambda > 1$ corresponds to a density decrease and $\lambda < 1$ to a density increase. This form of scale invariance is exact only for systems with Euler-homogeneous interactions (plus a constant)[13]. It is a good approximation, however, for the condensed phases of many systems in which this property is not obvious from inspection of the analytical expression for $U(\mathbf{R})$, thus the term 'hidden scale invariance'[39–42]. Equation (1), which is formally equivalent to the conformal-invariance condition $U(\mathbf{R}_a) = U(\mathbf{R}_b) \Leftrightarrow U(\lambda\mathbf{R}_a) = U(\lambda\mathbf{R}_b)$, implies invariance of structure and dynamics along the configurational adiabats in the phase diagram[39]. These lines are referred to as isomorphs[42]. It was very recently shown by Maimbourg and Kurchan[43] that in high dimensions all pair-potential systems obey hidden scale invariance in their condensed phase. Experimentally, hidden scale invariance has been demonstrated directly and indirectly for molecular van der Waals bonded liquids and polymers[44–46]. Further evidence for the existence of isomorphs comes from computer simulations of single-component systems[40,42] as well as, for example, of glass-forming mixtures[47], nanoflows[48], molecular models[38] and molecular dynamics (MD) simulations of the dynamics of most metallic elements based on quantum-mechanical, density-functional-theory potentials[49]. Isomorphs have also been demonstrated in simulations of out-of-equilibrium situations like zero-temperature shear flows of glasses or nonlinear steady-state liquid flows (see, for example, ref. 38 and its references). It is important to emphasize, however, that not all condensed matter exhibits hidden scale invariance; for instance, water is a notable exception[41]. The general picture is that most metals and organic van der Waals bonded systems obey equation (1) to a good approximation in the condensed-phase part of their thermodynamic phase diagram, whereas systems with strong directional bonding generally do not[38]. The former systems are simpler than the latter because their phase diagrams are

effectively one-dimensional in regard to structure and dynamics, reminiscent of the hard-sphere system. Systems with hidden scale invariance are sometimes referred to as Roskilde (R) simple[35,50–62] to distinguish them from simple systems traditionally defined as pair-potential systems[16]. The theory presented below makes use of R simple systems' almost one-dimensional phase diagrams[38] and gives corrections to the hard-sphere picture of melting and freezing calculated by the first-order Taylor expansions. Figure 2 illustrates the idea.

Along an isomorph the structure is invariant in the reduced-unit system defined[42] by the length unit $\rho^{-1/3}$ ($\rho \equiv N/V$ is the number density and $V$ is the system volume), the energy unit $k_B T$ ($T$ is the temperature) and the time unit $\sqrt{m\rho^{-2/3}/k_B T}$ ($m$ is the particle mass). Figure 1b shows the LJ liquid's static structure factor $S(q)$ along an isomorph close to the freezing line (used below as the liquid-state reference isomorph) plotted for a range of temperatures. A comparison with Fig. 1a confirms the recent finding of Heyes and Branka[32] that the freezing line is not an exact isomorph, although it is close to one.

The melting pressure as a function of temperature, $p_m(T)$, can be predicted from information obtained at a single coexistence reference state point. The details about how this works are given in the 'Methods' section. The argument may be summarized as follows. Recalling that the entropy as a function of density and temperature is a sum of an ideal-gas term and an 'excess' term $S_{ex}$ (ref. 16), isomorphs are the phase-diagram lines of constant excess entropy for any system obeying equation (1)[39,42]. A computer simulation at the liquid/solid reference state point generates a series of configurations $\mathbf{R}_1^0, \ldots, \mathbf{R}_n^0$. Scaling each of these uniformly to density $\rho$ one obtains configurations representative for the state point with density $\rho$ and temperature $T$ on the isomorph through the reference state point[39] in which $T$ is identified from the configurational temperature expression[63] $k_B T = \langle (\nabla U)^2 \rangle / \langle \nabla^2 U \rangle$. The average potential energy $U$ and virial $W$ at the state point $(\rho, T)$ are likewise found by averaging over the scaled configurations. The key assumption here is that the canonical probabilities of the scaled configurations are identical to those of the original configurations, which follows from equation (1)[39] (thus no new MD simulations are required). As shown in the 'Methods' section, in conjunction with the excess isochoric specific heat $C_V^{ex}$ calculated from the potential-energy fluctuations of the scaled configurations ($C_V^{ex} = \langle (\Delta U)^2 \rangle / k_B T^2$) and the so-called density-scaling exponent $\gamma \equiv (\partial \ln T / \partial \ln \rho)_{S_{ex}}$ also calculated from the fluctuations ($\gamma = \langle \Delta U \Delta W \rangle / \langle (\Delta U)^2 \rangle$), one has enough information to determine the thermodynamics of freezing and melting, as well as the variation along the melting line of isomorph-invariant properties like the Lindemann ratio and the reduced-unit viscosity.

**The LJ system.** For LJ type systems, the general procedure described above may be implemented analytically by making use of the fact that because the structure is isomorph invariant, it is possible to calculate the variation of the average potential energy and other relevant quantities analytically along an isomorph. This is done as follows. In reduced coordinates the pair correlation function $g(\tilde{r})$ is isomorph invariant ($\tilde{r} = \rho^{1/3} r$). Consequently, for pairs of LJ particles at distance $r$ the thermal average $\langle r^{-n} \rangle$ scales with density as $\rho^{n/3}$ along an isomorph. Thus $\langle r^{-n} \rangle \propto \rho^{n/3}$ with a proportionality constant that only depends on $S_{ex}$, implying that the average potential energy $U$ is of the form[64] $U = A_{12}(S_{ex})\tilde{\rho}^4 + A_6(S_{ex})\tilde{\rho}^2$ in which $\tilde{\rho}$ is the density relative to the reference state-point density and $A_6(S_{ex}) < 0$ derives from the attractive term of the LJ potential. Since $T = (\partial U/\partial S_{ex})_\rho$, one has $T = A'_{12}(S_{ex})\tilde{\rho}^4 + A'_6(S_{ex})\tilde{\rho}^2$. It follows that if the five quantities $S_{ex}$, $A_{12}(S_{ex})$, $A_6(S_{ex})$, $A'_{12}(S_{ex})$ and

$A'_6(S_{ex})$ are known, the excess Helmholtz free energy, $U-TS_{ex}$, is known along the reference isomorph. The required quantities are easily determined from reference state-point simulations (see the 'Methods' section)—for instance the reference state-point's potential energy and virial give two linear equations for determining $A_{12}(S_{ex})$ and $A_6(S_{ex})$. Once the excess Helmholtz free energy is known along the reference isomorph, the Gibbs free energy is found by adding the ideal-gas Helmholtz free energy and the $pV$ term ($pV = Nk_BT + W$ in which the virial is given[42] by $W = (\partial U/\partial \ln \tilde{\rho})_{S_{ex}} = 4A_{12}(S_{ex})\tilde{\rho}^4 + 2A_6(S_{ex})\tilde{\rho}^2$).

**Comparing theory to simulation results for the LJ system.** Following the above procedure, we generated two reference isomorphs for the LJ system starting from the coexistence state point with temperature $T_0 = 2.0\varepsilon/k_B$, a liquid-phase isomorph and a crystal-phase isomorph. Gibbs free energy of the liquid phase at coexistence, $G_l(T)$, is found by utilizing the fact that the freezing line is close to an isomorph. Since $(\partial G/\partial p)_T = V$, a good approximation to $G_l$ at coexistence is

$$G_l(p_m(T), T) \cong G_l^I(T) + V_l^I(T)(p_m(T) - p_l^I(T)). \quad (2)$$

Here, $p_m(T)$ is the coexistence pressure to be determined; $G_l^I(T)$ is the Gibbs-free energy, $V_l^I(T)$ the volume and $p_l^I(T)$ the pressure along the liquid-state reference isomorph. These quantities are all known functions of the (relative) density on the isomorph

henceforth denoted by $\tilde{\rho}^I$, which for temperature $T$ is found by solving $T = A'_{12}(S_{ex})(\tilde{\rho}^I)^4 + A'_6(S_{ex})(\tilde{\rho}^I)^2$.

An analogous expression applies for the crystal's Gibbs free energy, of course, again involving only parameters determined from reference state-point simulations. The coexistence pressure is determined by equating the liquid and solid phases' Gibbs free energies. As shown in the 'Methods' section (equation (21)), this results in $p_m(T)(V_l^I(T) - V_s^I(T)) = C_1(T) + C_2(T) - C_3(T)$ in which $C_1(T)$ is the difference between $U_s^I(T) - (T/T_0)U_s^I(T_0)$ and the analogous term for the liquid reference isomorph (here $U_s^I(T)$ is the crystal's potential energy along the reference isomorph), $C_2(T)$ is the difference between $Nk_BT\ln(\tilde{\rho}_s^I(T))$ and the analogous liquid term and $C_3(T)$ is the difference between $(T/T_0)W_s^I(T_0)$ and the analogous liquid term.

Figure 3a,b compare the theoretically predicted $p_m(T)$ to the coexistence pressure computed numerically by means of the interface-pinning method[37]. The density of the crystalline and liquid phases may also be computed by means of a first-order Taylor expansion working from the reference isomorph (see the 'Methods' section). Figure 3c compares the predicted $(\rho,T)$ phase diagram based on equation (26) to that obtained by the interface-pinning MD simulations. Finally, Fig. 3d shows the predicted and simulated fusion entropy $\Delta S_{fus}$ and enthalpy $\Delta H_{fus}$, the latter quantity being of course measured in experiments as the latent heat. In all cases there is good agreement between theoretical prediction and simulations.

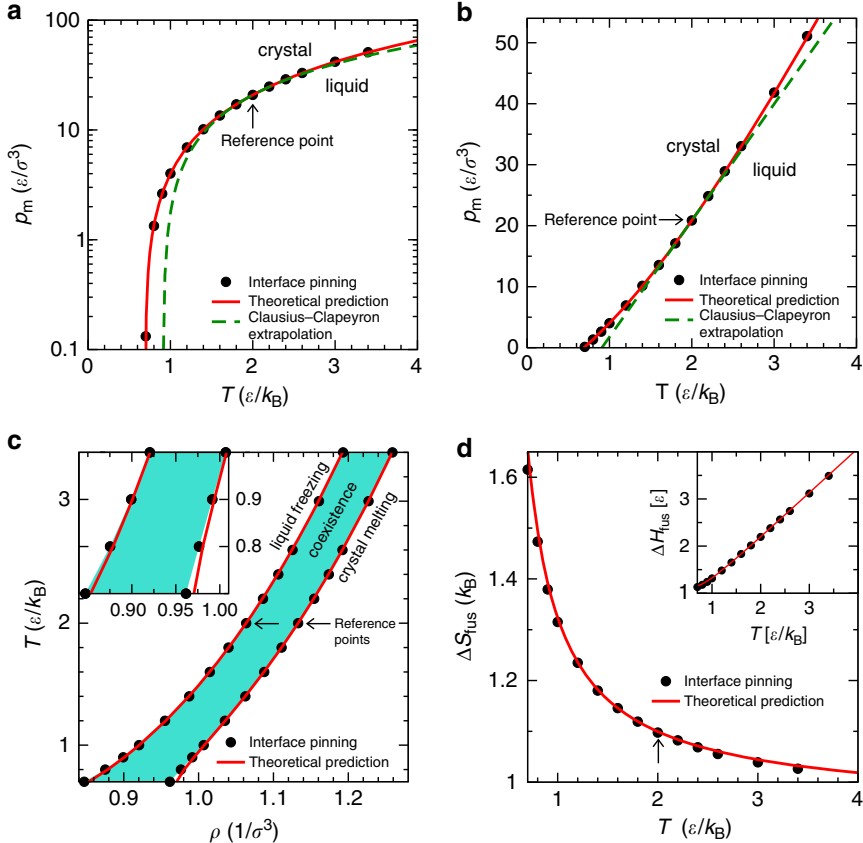

**Figure 3 | Theoretical predictions (full red curves) and results of MD simulations (black dots) for the LJ system.** The theoretical predictions are based on simulations at the coexistence reference state point indicated by an arrow in each figure ($T_0 = 2.0\varepsilon/k_B$), the MD simulations employed the interface-pinning method[37], see the 'Methods' section. No fitting was done in these figures—the only input to the theory is properties of the coexisting liquid and crystal at the reference temperature. (**a**) Temperature–pressure phase diagram. The green dashed line marks the expectation based on a linear extrapolation of the Clausius–Clapeyron relation $dp_m/dT = \Delta S_{fus}/\Delta V_{fus}$ from the reference state point, that is, assuming that the entropy of fusion and the volume change are both constant. (**b**) The same data plotted with a linear pressure axis. (**c**) The freezing and melting lines in the density–temperature phase diagram; the coloured area is the coexistence region. (**d**) Fusion entropy (main panel) and enthalpy (inset).

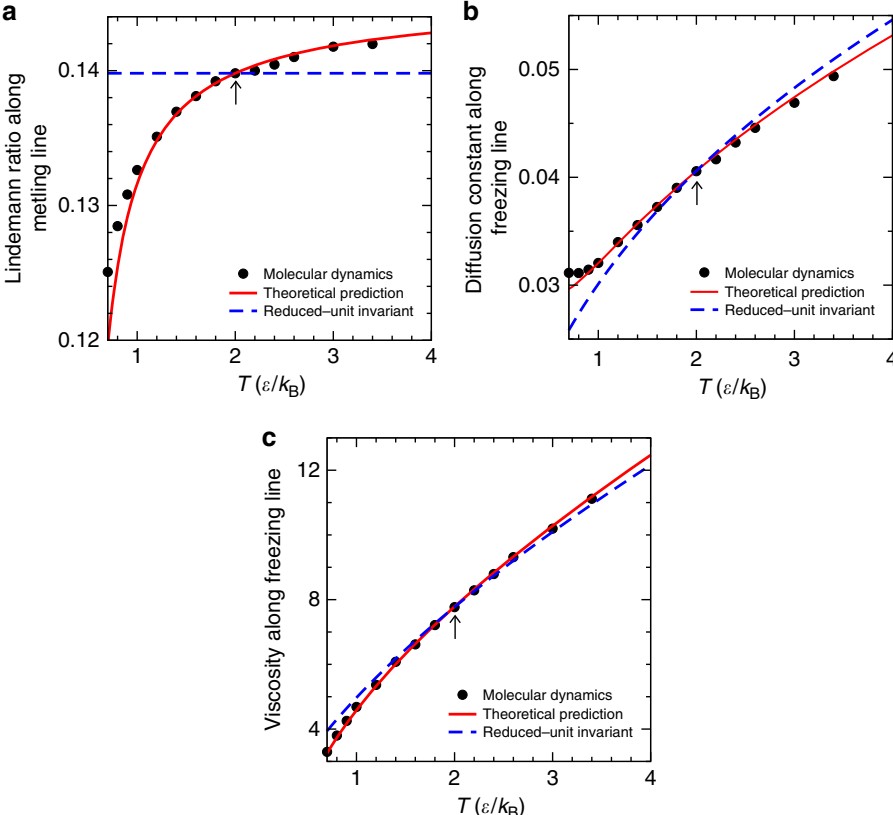

**Figure 4 | Predictions (red curves) versus results of computer simulations (black dots) for three properties along the freezing/melting lines of the LJ system.** The blue dashed lines show the predictions if perfect invariance of structure and dynamics in reduced units applies along the freezing/melting lines, the arrows indicate the reference state point upon which the predictions are based. (**a**) Lindemann ratio along the melting line. (**b**) Self-diffusion constant along the freezing line. (**c**) Viscosity along the freezing line.

Having in mind the fact that the pressure at the triple point is very low for the LJ system, we estimate the triple point temperature to $T_{\text{tp}} = 0.688(2)\varepsilon/k_{\text{B}}$ from the theory by assuming zero pressure. This is within the statistical uncertainty of the triple point temperature computed with the interface-pinning method. For comparison, a linear extrapolation of the Clausius–Clapeyron relation from the reference temperature (the green dashed lines in Fig. 3a,b) predicts a triple point temperature of $0.909(2)\varepsilon/k_{\text{B}}$.

Since the melting line is not an isomorph, the Lindemann ratio is not invariant along it. The theory estimates the deviation from a constant Lindemann ratio by a first-order Taylor expansion from the reference isomorph (see Fig. 2 and the 'Methods' section). Figure 4a demonstrates good, though not perfect agreement between theory and numerical computations of the Lindemann ratio. The liquids' self-diffusion constant plays an important role for the crystal growth rate as expressed, for example, in the Wilson–Frenkel law[65,66]. This motivated us to use the theory also for calculating the liquid's diffusion constant variation along the freezing line (Fig. 4b). Another important component for crystal growth is the thermodynamic driving force on the crystal–liquid interface, which is the Gibbs free energy difference between the two phases, $\Delta G \cong (T_{\text{m}} - T)\Delta S_{\text{fus}}$ ($\Delta S_{\text{fus}}$ is shown on Fig. 3d). Finally, Fig. 4c shows the viscosity along the freezing line. In all cases the blue dashed lines mark the prediction if the dynamics were invariant in reduced units, that is, if the freezing/melting lines were isomorphs.

## Discussion
The theory presented above predicts the thermodynamics of freezing and melting from a single coexistence state point. The

theory also enables one to calculate the deviations from the invariance of several quantities along the melting line predicted by the hard-sphere melting picture[16,22,30–32]. The theory is analytic for LJ type systems, that is, systems involving a pair potential that is a difference of two inverse power laws, but the framework developed applies to any system with hidden scale invariance, including molecular systems. The theory works well for the LJ system, with the largest deviations found close to the triple point where the structure is less invariant along the reference isomorph (Fig. 1b).

Having established a firm foundation for the thermodynamics of freezing and melting for R simple systems, it is our hope that it will soon be possible to address the exciting questions of how nucleation and growth proceed, processes that are not well understood even for simple systems beyond the hard-sphere system[67]. It seems likely that variations of the nucleation and growth mechanisms along the melting line can be analyzed in the same way as above, that is, by utilizing the fact that the freezing and melting lines are close to isomorphs along which the dynamics is invariant to a quite good approximation.

It is not clear to which degree this approach to melting can be generalized to quantum systems for which an outstanding question is the possible existence of a zero-temperature quantum fluid of metallic hydrogen. The quantum nature of the proton modifies classical melting, for example by increasing the value of the Lindemann ratio[68]. It would be interesting to investigate whether melting of quantum crystals may be understood in the above framework, but this awaits the development of an isomorph theory for quantum systems. In ongoing work we are addressing another open question, namely whether the above can

**Table 1 | Quantities characterizing the two reference state points in coexistence.**

| $T_0 = 2.0\ \varepsilon/k_B$ | Liquid | Crystal |
|---|---|---|
| $V_0/N\ [\sigma^3]$ | 0.9403(2)* | 0.8827(2) |
| $\rho_0\ [\sigma^{-3}]$ | 1.0633(2) | 1.1329(2) |
| $U_0/N\ [\varepsilon]$ | $-4.7792(2)$ | $-5.7774(2)$ |
| $W_0/N\ [\varepsilon]$ | 17.5418(7) | 16.3628(6) |
| $\gamma_0$ | 4.9164(8) | 4.8704(8) |
| $C^{ex}_{V,0}/N\ [k_B]$ | 1.323(5) | 1.301(7) |
| $B_0 N\ [\varepsilon/k_B^2]$ | 6.9(5) | 7.2(5) |
| $L_0$ | — | 0.1398(2) |
| $(\partial L/\partial T)_\rho\ [k_B/\varepsilon]$ | — | 0.041(2) |
| $\tilde{D}_0$ | 0.02921(9) | — |
| $(\partial\tilde{D}/\partial T)_\rho\ [k_B/\varepsilon]$ | 0.0201(4) | — |
| $\tilde{\eta}_0$ | 5.2487(6) | — |
| $(\partial\tilde{\eta}/\partial T)_\rho\ [k_B/\varepsilon]$ | $-2.60(14)$ | — |

These numbers were used for calculating the theoretical predictions in Figs 3 and 4 (red curves). *Numbers in parenthesis give the estimated statistical uncertainty using a 95% confidence interval.

be generalized to deal with more realistic systems, for instance metals for which density-functional-theory computer simulations nowadays give realistic representations of the physics and have demonstrated hidden scale invariance for most metals[49].

## Methods

**Computer simulations.** We studied a LJ system of $N = 5,000$ particles with pair interactions truncated and shifted at $6\sigma$. Coexistence pressures, $p_m$, are computed with the interface-pinning method[37] in which coexistence state points are determined by computing the thermodynamic driving force on a solid-liquid interface. Table 1 lists the energy $U_0$ and virial $W_0$ at the reference temperature $T_0 = 2\varepsilon/k_B$ for both the liquid and crystal states at coexistence. The $A_{12}$ and $A_6$ coefficients (for the liquid and the crystal separately) are computed from reference state-point data using equation (8) below. The derivatives of the $A$ coefficients with respect to excess entropy, $A'_{12}$ and $A'_6$, are computed from reference state-point data using equation (11) with the $\gamma_0$'s listed in Table 1. Melting pressures (Fig. 3a,b) are computed from reference state-point data using equation (21) in which the potential energies along the two reference isomorphs are expressed in equation (6). The densities along the liquid and crystal reference isomorphs are found as functions of temperature by inversion of equation (9) (upper equation). The second derivatives of the $A$ coefficients, $A''_{12}$ and $A''_6$, are given by equation (15) where the reference state point excess heat capacity $C^{ex}_{V,0}$ and $B_0 \equiv (\partial(T/C^{ex}_V)/\partial\ln\rho)_{S_{ex}}$ are listed in Table 1. The freezing and melting densities (Fig. 3c) are computed from the pressures by combining equations (22) and (25). The entropy of fusion $\Delta S_{fus}$ (Fig. 3d) is computed by combining equations (27–30). The value of the Lindemann ratio $L$ of the crystal at the reference temperature, $L_0$ and its temperature derivative along an isochore, $(\partial L/\partial T)_\rho$, are listed in Table 1. By letting $X = L$ in equations (32) and (38), we arrive at the prediction shown in Fig. 4a. Similarly, the predictions of the self-diffusion constant $D$ (Fig. 4b) and viscosity $\eta$ (Fig. 4c) are found by letting $X = \tilde{D} = D\rho^{1/3}\sqrt{m/k_BT}$ and $X = \tilde{\eta} = \eta/\rho^{2/3}\sqrt{mk_BT}$, respectively. $D$ is determined from the long-time limit of the mean-square displacement; $\eta$ is computed using the SLLOD algorithm as detailed in ref. 27 except that in the present work we increased the number of particles to 4,096 and used the above-mentioned larger cutoff.

We proceed to describe the theory in detail. The reference state point is selected at coexistence, that is, with known temperature $T_0$ and pressure $p_0$. There are two different reference densities, a solid and a liquid one, below denoted, respectively, by $\rho_{s,0}$ and by $\rho_{l,0}$. In the density–temperature phase diagram there are two reference isomorphs. The arguments developed in the next two sections refer to either one of these.

**Isomorph characteristics of arbitrary R simple systems.** As mentioned, the temperature–pressure reference state point defines two reference density–temperature state points, a liquid and a solid one. Let us focus on one of these with density $\rho_0$ and temperature $T_0$ (thus dropping in this and the next subsection subscripts $s$ and $l$). From an NVT MD equilibrium simulation (for example, with a Nosé–Hoover thermostat) $n$ configurations $\mathbf{R}^0_1, \mathbf{R}^0_2, \ldots, \mathbf{R}^0_n$ are sampled. In order to map out the reference isomorph parametrized by density, one first identifies the temperature $T$ such that $(\rho, T)$ is on the isomorph through the reference state point $(\rho_0, T_0)$. This is done as follows. If the configurations scaled uniformly to density $\rho$ are denoted by $\mathbf{R}_1, \mathbf{R}_2, \ldots, \mathbf{R}_n$ in which $\mathbf{R}_i = (\rho_0/\rho)^{1/3}\mathbf{R}^0_i$, the temperature $T$ is determined from the standard configurational temperature expression (in which the averages are over the $n$ sampled configurations)

$$k_B T = \frac{\langle(\nabla U(\mathbf{R}_i))^2\rangle_i}{\langle\nabla^2 U(\mathbf{R}_i)\rangle_i}. \tag{3}$$

This determines the function $T(\tilde{\rho}^I)$ where we define the relative density along the isomorph by $\tilde{\rho}^I \equiv \rho^I/\rho_0$ with superscript $I$ indicating 'isomorph' (thus $T(1) = T_0$). By averaging the potential energy $U(\mathbf{R})$ and the virial $W(\mathbf{R}) \equiv (-1/3)\mathbf{R}\cdot\nabla U(\mathbf{R})$ over the scaled configurations one identifies the functions $U(\tilde{\rho}^I)$ and $W(\tilde{\rho}^I)$. $C^{ex}_V(\tilde{\rho}^I)$ is found from the scaled configurations' potential energy via $C^{ex}_V = \langle(\Delta U)^2\rangle/k_BT^2$ in which $T = T(\tilde{\rho}^I)$. The density-scaling exponent $\gamma(\tilde{\rho}^I) \equiv (\partial\ln T/\partial\ln\rho)_{S_{ex}}$ may be found either via the statistical-mechanical identity[42,69] $\gamma = \langle\Delta U\Delta W\rangle/\langle(\Delta U)^2\rangle$ or simply by taking the derivative of an analytical approximation to the function $T(\tilde{\rho})$.

As shown in the below subsection 'The melting-line pressure', one now has enough information to calculate the pressure along the melting line, $p_m(T)$. To calculate the liquid and solid densities along the melting line (see subsection 'The freezing- and melting-line densities' below) one needs to know the below three partial derivatives. Denoting the derivative of the virial along the isomorph with respect to $\tilde{\rho}^I$ by $W'(\tilde{\rho}^I)$ and recalling that $W = (\partial U/\partial\ln\tilde{\rho})_{S_{ex}}$ and $T = (\partial U/\partial S_{ex})_{\tilde{\rho}}$ (refs 42,69), the three required quantities are given by

$$\left(\frac{\partial W}{\partial\ln\tilde{\rho}}\right)^I_{S_{ex}} = \tilde{\rho}^I W'(\tilde{\rho}^I)$$

$$\left(\frac{\partial W}{\partial S_{ex}}\right)^I_{\tilde{\rho}} = \frac{\partial^2 U}{\partial S_{ex}\partial\ln\tilde{\rho}} = \left(\frac{\partial T}{\partial\ln\tilde{\rho}}\right)_{S_{ex}} = T(\tilde{\rho}^I)\gamma(\tilde{\rho}^I) \tag{4}$$

$$\left(\frac{\partial S_{ex}}{\partial\ln\tilde{\rho}}\right)^I_T = -\frac{\left(\frac{\partial T}{\partial\ln\tilde{\rho}}\right)_{S_{ex}}}{\left(\frac{\partial T}{\partial S_{ex}}\right)_{\tilde{\rho}}} = -\frac{T(\tilde{\rho}^I)\gamma(\tilde{\rho}^I)}{T(\tilde{\rho}^I)/C^{ex}_V(\tilde{\rho}^I)} = -C^{ex}_V(\tilde{\rho}^I)\gamma(\tilde{\rho}^I).$$

The entropy of fusion $\Delta S_{fus}$ is calculated by use of equations (27–30) below. The three quantities needed here are given by

$$\left(\frac{\partial U}{\partial\ln\tilde{\rho}}\right)^I_{S_{ex}} = W(\tilde{\rho}^I)$$

$$\left(\frac{\partial U}{\partial S_{ex}}\right)^I_{\tilde{\rho}} = T(\tilde{\rho}^I) \tag{5}$$

$$\left(\frac{\partial S_{ex}}{\partial\ln\tilde{\rho}}\right)^I_T = -C^{ex}_V(\tilde{\rho}^I)\gamma(\tilde{\rho}^I).$$

**Isomorph characteristics of generalized LJ pair potentials.** The above quantities may be calculated analytically for generalized LJ pair potentials, that is, for systems of particles interacting via pair potential(s) given as a sum or difference of two inverse power laws, $r^{-m}$ and $r^{-n}$. The derivation given below applies for any exponents $m > n > 0$ and for general multi-component systems; its subsequent application to freezing and melting deals with single-component systems only.

Invariance of the structure along an isomorph implies that the thermodynamic average potential energy at a given state point, $U$, may be written $U = A_m\tilde{\rho}^{m/3} + A_n\tilde{\rho}^{n/3}$ (in this section the superscript $I$ is dropped on the reference isomorph density) in which the two $A$ coefficients are functions only of the excess entropy $S_{ex}$. For simplicity of notation we shall not indicate the $S_{ex}$ dependence. The first and second order derivatives of $A_m$ with respect to $S_{ex}$ are marked by $A'_m$ and $A''_m$ and likewise for $A_n$.

The identity for the virial $W = (\partial U/\partial\ln\tilde{\rho})_{S_{ex}}$ implies

$$U = A_m\tilde{\rho}^{m/3} + A_n\tilde{\rho}^{n/3}$$
$$W = \frac{m}{3}A_m\tilde{\rho}^{m/3} + \frac{n}{3}A_n\tilde{\rho}^{n/3}. \tag{6}$$

At the reference state point $\tilde{\rho} = 1$, so for determining $A_m$ and $A_n$ from reference state-point data we have the following two equations:

$$U_0 = A_m + A_n$$
$$W_0 = \frac{m}{3}A_m + \frac{n}{3}A_n. \tag{7}$$

This implies

$$A_m = \frac{3W_0 - nU_0}{m - n}$$
$$A_n = \frac{mU_0 - 3W_0}{m - n}. \tag{8}$$

From the identity $T = (\partial U/\partial S_{ex})_{\tilde{\rho}}$ and the definition of the density-scaling exponent, $\gamma \equiv (\partial\ln T/\partial\ln\tilde{\rho})_{S_{ex}}$, we get

$$T = A'_m\tilde{\rho}^{m/3} + A'_n\tilde{\rho}^{n/3}$$
$$\gamma T = \frac{m}{3}A'_m\tilde{\rho}^{m/3} + \frac{n}{3}A'_n\tilde{\rho}^{n/3}. \tag{9}$$

For determining $A'_m$ and $A'_n$ from reference state-point data one has

$$T_0 = A'_m + A'_n$$
$$\gamma_0 T_0 = \frac{m}{3}A'_m + \frac{n}{3}A'_n. \tag{10}$$

This implies

$$A'_m = \frac{3\gamma_0 - n}{m - n} T_0$$

$$A'_n = \frac{m - 3\gamma_0}{m - n} T_0. \tag{11}$$

In order to arrive at equations for $A''_m$ and $A''_n$, we first note that $C_V^{ex} = T(\partial S_{ex}/\partial T)_{\tilde\rho}$, that is, $(\partial T/\partial S_{ex})_{\tilde\rho} = T/C_V^{ex}$. This implies that $T/C_V^{ex} = A''_m \tilde\rho^{m/3} + A''_n \tilde\rho^{n/3}$. If we define a thermodynamic quantity $B$ by

$$B \equiv \left(\frac{\partial(T/C_V^{ex})}{\partial\ln\tilde\rho}\right)_{S_{ex}}, \tag{12}$$

one has

$$\frac{T}{C_V^{ex}} = A''_m \tilde\rho^{m/3} + A''_n \tilde\rho^{n/3}$$

$$B = \frac{m}{3} A''_m \tilde\rho^{m/3} + \frac{n}{3} A''_n \tilde\rho^{n/3}. \tag{13}$$

The two equations for determining $A''_m$ and $A''_n$ from reference state-point data are thus

$$\frac{T_0}{C_{V,0}^{ex}} = A''_m + A''_n$$

$$B_0 = \frac{m}{3} A''_m + \frac{n}{3} A''_n. \tag{14}$$

This implies

$$A''_m = \frac{3B_0 - nT_0/C_{V,0}^{ex}}{m - n}$$

$$A''_n = \frac{mT_0/C_{V,0}^{ex} - 3B_0}{m - n}. \tag{15}$$

In summary, we have shown that for each of the two reference isomorphs the six numbers $A_m$, $A_n$, $A'_m$, $A'_n$, $A''_m$ and $A''_n$ may be found from reference state-point simulations determining: (1) the potential energy $U_0$, (2) the virial $W_0$, (3) the temperature $T_0$, (4) the excess isochoric specific heat $C_{V,0}^{ex}$, (5) the density-scaling exponent $\gamma_0$ and (6) the derivative of $C_V^{ex}$ along the isomorph via the quantity $B_0$ defined in equation (12). The first three quantities are determined directly. The next two quantities are determined from fluctuations at the reference state point: $C_{V,0}^{ex} = \langle(\Delta U)^2\rangle/k_B T_0^2$ and $\gamma_0 = \langle\Delta W\Delta U\rangle/\langle(\Delta U)^2\rangle$. Finally, the quantity $B_0$ is most accurately found from simulations along the reference isomorph carried out close to the reference state point, although in principle $B_0$ can be calculated from fluctuations at the reference state point (those needed are of third order and consequently of considerable numerical uncertainty). We calculated $B_0$ numerically by directly applying equation (12); alternatively, following the methods used in ref. 70 one may rewrite $B$ as $B = (\gamma T/C_V^{ex})[1 + (\partial\ln\gamma/\partial\ln T)_\rho]$ and evaluate $B_0$ from the (rather weak) constant-density temperature variation of $\gamma$ at the reference state point.

**The melting-line pressure.** In the temperature–pressure phase diagram the freezing and melting lines are identical. This section shows how to calculate the pressure on this line as a function of temperature, $p_m(T)$, which is determined by equating the liquid and solid phase's Gibbs free energies. Recalling that $V = (\partial G/\partial p)_T$ we estimate these from the Gibbs free energies along the isomorphs, $G_l^I(T)$ and $G_s^I(T)$, as follows (below $F_l^I(T)$ is the Helmholtz free energy along the liquid reference isomorph and likewise for the solid)

$$G_l(T, p_m(T)) \cong G_l^I(T) + V_l^I(T)(p_m(T) - p_l^I(T)) = F_l^I(T) + V_l^I(T)p_m(T)$$

$$G_s(T, p_m(T)) \cong G_s^I(T) + V_s^I(T)(p_m(T) - p_s^I(T)) = F_s^I(T) + V_s^I(T)p_m(T). \tag{16}$$

The coexistence condition $G_l(T, p_m) = G_s(T, p_m)$ leads to

$$p_m(T)(V_l^I(T) - V_s^I(T)) = F_s^I(T) - F_l^I(T). \tag{17}$$

If $F_{id}$ is the ideal-gas Helmholtz free energy, the Helmholtz free energy along the liquid isomorph is given by

$$F_l^I(T) = U_l^I(T) - TS_{ex,l}^I + F_{id}(T, \rho_l^I(T)). \tag{18}$$

An analogous expression applies for the solid isomorph's Helmholtz free energy, $F_s^I(T)$, of course. The two constants $S_{ex,l}^I$ and $S_{ex,s}^I$ are not known, but one needs only their difference. This is determined from the equilibrium condition at the reference state point, $G_l(T_0, p_0) = G_s(T_0, p_0)$ as expressed in equation (17), leading, since $pV = Nk_BT + W$ and $F_{id}(T, \rho_l) - F_{id}(T, \rho_s) = Nk_BT\ln(\rho_l/\rho_s)$, to

$$T_0(S_{ex,l}^I - S_{ex,s}^I) = (U_{l,0} - U_{s,0}) + Nk_BT_0\ln(\rho_{l,0}/\rho_{s,0}) + (W_{l,0} - W_{s,0}). \tag{19}$$

The coexistence condition, equation (17), thus becomes (dropping the explicit temperature dependencies)

$$p_m(V_l^I - V_s^I) = (U_s^I - U_l^I) - \frac{T}{T_0}((U_{s,0} - U_{l,0}) + Nk_BT_0\ln(\rho_{s,0}/\rho_{l,0}) + (W_{s,0} - W_{l,0})) + Nk_BT\ln(\rho_s^I/\rho_l^I) \tag{20}$$

or, in terms of the relative density along the respective isomorphs $\tilde\rho^I$,

$$p_m(V_l^I - V_s^I) = \left(U_s^I - \frac{T}{T_0} U_{s,0}\right) - \left(U_l^I - \frac{T}{T_0} U_{l,0}\right) + Nk_BT\ln(\tilde\rho_s^I/\tilde\rho_l^I) + \frac{T}{T_0}(W_{l,0} - W_{s,0}). \tag{21}$$

In the case of an arbitrary potential there is no analytical expression for the (average) potential energy as a function of density. Here, the density (of each phase) is the control parameter and $T$ is identified from equation (3), resulting by numerical inversion in two functions $\tilde\rho_s^I(T)$ and $\tilde\rho_l^I(T)$. In the case of generalized LJ pair potentials, for a given temperature $T$ the functions $\tilde\rho_l^I(T)$ and $\tilde\rho_s^I(T)$ are found by solving equation (9) (in general numerically, but analytically for the 12-6 LJ system), using the $A'$ coefficients of equation (11). The potential energy along the isomorphs is given by equation (6).

**The freezing- and melting-line densities.** We work from the respective reference isomorphs knowing as functions of temperature the coexistence pressure, and the pressure along the reference isomorphs. From this information one calculates the solid and liquid densities by moving on an isotherm from the reference isomorph to the freezing/melting line (Fig. 2). In both cases we define the isothermal difference $DW \equiv W(T) - W^I(T)$. Here and thoughout the paper D refers to isothermal differences between the reference isomorph and the freezing/melting line.

At any given temperature $T$ the density $\tilde\rho$ of the liquid/solid at coexistence is calculated from

$$DW \cong \left(\frac{\partial W}{\partial\ln\tilde\rho}\right)_T^I D\ln\tilde\rho = \left(\frac{\partial W}{\partial\ln\tilde\rho}\right)_T^I \ln(\tilde\rho/\tilde\rho^I). \tag{22}$$

If $(\partial W/\partial\ln\tilde\rho)_T^I$ is known, we can determine $\tilde\rho$ from equation (22).

The following standard identity is used

$$\left(\frac{\partial W}{\partial\ln\tilde\rho}\right)_T = \left(\frac{\partial W}{\partial\ln\tilde\rho}\right)_{S_{ex}} + \left(\frac{\partial W}{\partial S_{ex}}\right)_{\tilde\rho}\left(\frac{\partial S_{ex}}{\partial\ln\tilde\rho}\right)_T. \tag{23}$$

In the case of an arbitrary potential, the three terms on the right hand side are calculated from equation (4). For the generalized LJ case, these terms are expressed in terms of the $A$ coefficients by making use of equations (6) and (9), resulting in

$$\left(\frac{\partial W}{\partial\ln\tilde\rho}\right)_{S_{ex}}^I = \left(\frac{m}{3}\right)^2 A_m(\tilde\rho^I)^{m/3} + \left(\frac{n}{3}\right)^2 A_n(\tilde\rho^I)^{n/3}$$

$$\left(\frac{\partial W}{\partial S_{ex}}\right)_{\tilde\rho}^I = \frac{m}{3} A'_m(\tilde\rho^I)^{m/3} + \frac{n}{3} A_n(\tilde\rho^I)^{n/3}$$

$$\left(\frac{\partial S_{ex}}{\partial\ln\tilde\rho}\right)_T^I = -\frac{\left(\frac{\partial T}{\partial\ln\tilde\rho}\right)_{S_{ex}}^I}{\left(\frac{\partial T}{\partial S_{ex}}\right)_{\tilde\rho}^I} = -\frac{\frac{m}{3}A'_m(\tilde\rho^I)^{m/3} + \frac{n}{3}A'_n(\tilde\rho^I)^{n/3}}{A''_m(\tilde\rho^I)^{m/3} + A''_n(\tilde\rho^I)^{n/3}}. \tag{24}$$

We thus have in the generalized LJ case

$$\left(\frac{\partial W}{\partial\ln\tilde\rho}\right)_T^I = \left(\frac{m}{3}\right)^2 A_m(\tilde\rho^I)^{m/3} + \left(\frac{n}{3}\right)^2 A_n(\tilde\rho^I)^{n/3} - \frac{\left(\frac{m}{3}A'_m(\tilde\rho^I)^{m/3} + \frac{n}{3}A'_n(\tilde\rho^I)^{n/3}\right)^2}{A''_m(\tilde\rho^I)^{m/3} + A''_n(\tilde\rho^I)^{n/3}}. \tag{25}$$

In both the arbitrary potential case and that of generalized LJ systems, the equation for the density $\rho(T) = N/V(T)$ is found from equation (22) solved numerically in the form

$$p_m(T)V(T) - Nk_BT - W^I(T) = \left(\frac{\partial W}{\partial\ln\tilde\rho}\right)_T^I \ln(\tilde\rho/\tilde\rho^I). \tag{26}$$

**The entropy of fusion.** In this section we calculate the constant-pressure entropy of fusion $\Delta S_{fus}$. One way to do this is to use the Clausius–Clapeyron equation $dp_m/dT = \Delta S_{fus}/\Delta V_{fus}$ in which we now know all quantities except $\Delta S_{fus}$. An alternative method similar to the above proceeds as follows. Across the melting line one has $\Delta G_{fus} = 0$, that is, $\Delta E_{fus} - T\Delta S_{fus} + p_m\Delta V_{fus} = 0$ ($E$ is the total energy). Since the kinetic energy is the same for liquid and solid at the given temperature $T$, one has $\Delta E_{fus} = \Delta U_{fus}$ and thus

$$\Delta S_{fus} = \frac{\Delta U_{fus} + p_m\Delta V_{fus}}{T}. \tag{27}$$

This equation is used for evaluating $\Delta S_{fus}$ from interface-pinning simulations. It is also used for predicting $\Delta S_{fus}(T)$ by proceeding as follows. We have predictions for $p_m = p_m(T)$ and for $\Delta V_{fus} = V_l(T) - V_s(T)$. The missing term is $\Delta U_{fus} = \Delta U_{fus}(T)$, which is estimated via

$$\Delta U_{fus} = U_l^I(T) + \left(\frac{\partial U}{\partial\ln\tilde\rho}\right)_T^{I,l}\ln(\tilde\rho_l(T)/\tilde\rho_l^I(T)) - U_s^I(T) - \left(\frac{\partial U}{\partial\ln\tilde\rho}\right)_T^{I,s}\ln(\tilde\rho_s(T)/\tilde\rho_s^I(T)). \tag{28}$$

The partial derivatives refer to the respective reference isomorph as in the last section, and these are evaluated like those of $W$. Thus,

$$\left(\frac{\partial U}{\partial\ln\tilde\rho}\right)_T^I = \left(\frac{\partial U}{\partial\ln\tilde\rho}\right)_{S_{ex}}^I + \left(\frac{\partial U}{\partial S_{ex}}\right)_{\tilde\rho}^I\left(\frac{\partial S_{ex}}{\partial\ln\tilde\rho}\right)_T^I. \tag{29}$$

In the case of an arbitrary potential, the three terms on the right hand side are calculated from equation (5). For the generalized LJ case, these terms may be expressed in terms of the $A$ coefficients of the reference isomorph as follows

$$\left(\frac{\partial U}{\partial \ln \tilde{\rho}}\right)_{S_{ex}}^{I} = W^{I} = \frac{m}{3} A_m (\tilde{\rho}^{I})^{m/3} + \frac{n}{3} A_n (\tilde{\rho}^{I})^{n/3}$$

$$\left(\frac{\partial U}{\partial S_{ex}}\right)_{\tilde{\rho}}^{I} = T = A'_m (\tilde{\rho}^{I})^{m/3} + A'_n (\tilde{\rho}^{I})^{n/3} \tag{30}$$

$$\left(\frac{\partial S_{ex}}{\partial \ln \tilde{\rho}}\right)_{T}^{I} = -\frac{\left(\frac{\partial T}{\partial \ln \tilde{\rho}}\right)_{S_{ex}}^{I}}{\left(\frac{\partial T}{\partial S_{ex}}\right)_{\tilde{\rho}}^{I}} = -\frac{\frac{m}{3} A'_m (\tilde{\rho}^{I})^{m/3} + \frac{n}{3} A_{n'} (\tilde{\rho}^{I})^{n/3}}{A''_m (\tilde{\rho}^{I})^{m/3} + A''_n (\tilde{\rho}^{I})^{n/3}}.$$

We now have all information required for calculating the entropy of fusion.

**Melting-line variation of isomorph invariants.** We finally turn to the problem of evaluating how much an isomorph invariant $X$—*in casu* the reduced vibrational crystalline mean-square displacement, the reduced liquid-state diffusion constant, and the reduced liquid-state viscosity—varies along the freezing/melting line. The starting point is that

$$X = \phi(S_{ex}). \tag{31}$$

On the one hand

$$\left(\frac{\partial X}{\partial T}\right)_\rho = \phi'(S_{ex}) \left(\frac{\partial S_{ex}}{\partial T}\right)_\rho = \phi'(S_{ex}) \frac{C_V^{ex}}{T}. \tag{32}$$

On the other hand we have the standard fluctuation formula

$$\left(\frac{\partial X}{\partial T}\right)_\rho = \frac{\langle \Delta X \Delta U \rangle}{k_B T^2}. \tag{33}$$

Combining these equations at the reference state point leads to (where subscript 0 denotes an equilibrium average at the reference state point)

$$\phi'(S_{ex}) = \frac{\langle \Delta X \Delta U \rangle_0}{k_B T_0 C_{V,0}^{ex}}. \tag{34}$$

Consider next an arbitrary temperature $T$ on the freezing/melting line. If $DS_{ex}$ is the difference between crystal (respectively) liquid excess entropy at melting and that of the corresponding reference isomorph at the same temperature and $D\rho$ likewise is the difference between crystal (respectively) liquid density at melting and that of the corresponding reference isomorph, we estimate $X$ via

$$X \cong X_0 + \phi'(S_{ex}) DS_{ex} \cong X_0 + \phi'(S_{ex}) \left(\frac{\partial S_{ex}}{\partial \rho}\right)_T D\rho. \tag{35}$$

Equation (4) implies

$$\left(\frac{\partial S_{ex}}{\partial \rho}\right)_T^I = -\frac{\gamma C_V^{ex}}{\rho}. \tag{36}$$

Thus we have

$$X \cong X_0 - \phi'(S_{ex}) \gamma C_V^{ex} \frac{D\rho}{\rho}. \tag{37}$$

This implies

$$X \cong X_0 - \left(\frac{\partial X}{\partial T}\right)_\rho \gamma T_0 \frac{C_V^{ex}}{C_{V,0}^{ex}} \frac{D\rho}{\rho} \tag{38}$$

in which the partial derivative is evaluated at the reference state point. If $X$ is a thermodynamic quantity, one may use the fluctuation expression, equation (34), to rewrite this as follows

$$X \cong X_0 - \frac{\langle \Delta X \Delta U \rangle_0}{k_B T_0} \gamma \frac{C_V^{ex}}{C_{V,0}^{ex}} \frac{D\rho}{\rho}. \tag{39}$$

This expression may be used in the case of an arbitrary potential, as well as for generalized LJ systems for which analytical expressions are available.

**Data availability.** The data presented in this study are available from the corresponding author upon request.

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

## Acknowledgements

We are indebted to Karolina Adrjanowicz and Kristine Niss for inspiring the present work via their attempts at interpreting crystallization data for van der Waals liquids in terms of the isomorph theory. This work was supported by the Villum Foundation by the YIP Grant VKR-023455 and by the Danish National Research Foundation by Grant DNRF61.

## Author contributions

The project was conceived by U.R.P. and N.P.B. Computations were carried out by U.R.P. and L.C. The theory was devised by J.C.D. based on initial works by N.P.B. and T.B.S. The paper was written by U.R.P. and J.C.D.

## Additional information

**Competing financial interests:** The authors declare no competing financial interests.

