## [Peer Review File · Nature Communications]

Reviewers' comments:

Reviewer #1 (Remarks to the Author):

The work deals with one of a most common and fundamental phenomenon encountered in the nature - the freezing/melting transition. This type of the phase transition has been a subject of intensive studies of many researches for decades and still remains unsolved problem.

The work develops the framework which provides the basis for calculating the fundamental thermodynamic properties of freezing and melting. The proposed method applies to the considerable class of systems and thus, contributes significantly to the basic research on the melting/freezing issue. It is based on the concept of the hidden scale-invariance and predicts the pressure, density and entropy along the melting line as well as the Lindemann ratio, viscosity and diffusion constant.

In my opinion the work can be of interest not only to the physicists and deserves to be published in Nature Communications.

The only my concern is the Supplementary Information which constitutes an essential part of the main text and in fact it is difficult to treat it as a supplementary material.

Some trivial algebra in the SI should be omitted (e.g., all the intermediate steps in getting eqs. 6,9,13.)

There are several compact phrases and jargon words which need to be changed or explained, some examples are listed blow.

p.2 "the canonical Lennard-Jones" - is too shortened and thus unclear

p.2 "longer ranged attractive London(van der Waals) forces" - London forces are part of the van der Waals forces

p.3 lambda in eq.(1) is to be explained

p.4 in Fig.1 caption as well as in the main text, "Structure" may need some extra comment what it denotes in this work.

p.4 the last sentence "The fact lies ... the melting is unique." is not obvious what the authors mean here.

p.2 in SI, "From an NVT equilibrium simulation" - a method which maintains constant T in calculations should be given.

p.2 in SI, the average $\langle \dots \rangle_i$ in eq.(1) is not defined; also, one can argue that the standard expression is the ratio of averages not the average of the ration,

p.3 in SI, do the identity for the virial W implies U in eq.(4)

p.6 "In the general case" - it is not obvious what "general case" is here mentioned

p.7 in SI, "A coefficients determined via eq.(9)" - in eq.(9) A' are determine; note in the text many times A instead of A' or A" are called

Reviewer #2 (Remarks to the Author):

A. Summary of key results

In this paper, the authors present a theory of thermodynamics and dynamics along the melting line of a simple model of monoatomic liquid, the Lennard-Jones system.

The theory allows one to compute many observables of interest, including the diffusion constant, the viscosity, the Lindemann ratio, as well as the transition line itself in the (P,T) and (ρ ,T) phase diagram.

B. Originality and interest

The paper is strongly based on previous results of the same authors about the concept of "hidden scale invariance" (e.g. refs. 38,39). I do not really see a major conceptual advance with respect to previous work.

C. Data & methodology

The theory seems robust, and the results are sound and accurate.

The presentation is clear, but in some points it is not completely clear what is a "rigorous" result and what is an assumption. For example, Eq.1 is stated to be mathematically equivalent to the same relation but with an equality. Is this really true? For what class of functions $U(R)$? And if it is true, wouldn't it be better to use directly the relation with equality in Eq.1, which would provide a rigorous justification for the scale invariance of dynamics and thermodynamics?

Another minor remark is that on page 2 the authors write "the Lindemann ratio is predicted to be universal...". But this is a trivial consequence of the Hard Sphere model, right? Since HS have a single melting point, the Lindemann ratio can only take a single value.

D. Statistics and uncertainties

This has been discussed correctly.

E. Conclusions

In conclusion, this is a nice and elegant work where the authors derive many properties of the Lennard-Jones system at melting from a set of simple assumptions and few numerical simulations. At the same time, the work looks like an "exercise" in the context of the general hidden scale invariance approach previously developed by the authors, rather than a major conceptual advance. For this reason, I am not sure that the paper meets the criteria of originality that are required by Nature Communication. The paper might be more appropriate for a specialised journal like J.Chem.Phys.

Reviewer #3 (Remarks to the Author):

The manuscript describes a new methodology for the theoretical and computational study of freezing and melting. The authors demonstrate that, for a large class of systems, it is possible to use the properties of a single state point to calculate a large range of properties related to melting at other state points. These properties include, for example, the entropy of fusion and the Lindemann ratio.

More concretely, the starting point of the present work are a class of systems exhibiting "hidden scale-invariance". These systems have been extensively studied in recent years, with many contributions from the authors of the present paper. The novel contribution of the present manuscript is to exploit the properties of this class of systems to propose a new theory that, from the properties at a particular state point, those at other state points can be calculated. This approach holds promise for significantly reducing the computational cost of the study of melting.

The theory is adequately presented, with some of the more technical details left for the supplementary, which allows for a good flow of the main manuscript. The validation and usefulness of the new theory is demonstrated using the Lennard-Jones system as a test case. The results are convincing, and should motivate further work to expand them to other, more realistic systems.

Overall, the manuscript is well-structured, the main ideas are presented in a coherent and clear manner, and the conclusions are strong. Nonetheless, I would like to make a few suggestions to the authors:

1. Regarding the realm of applicability of the theory, could the authors include a discussion on quantum melting? Some of the references provided by the authors are for high pressure research. In this field, one of the outstanding questions is the possible existence of a zero-temperature quantum fluid of metallic hydrogen. The quantum nature of the proton modifies classical melting, for example by increasing the value of the Lindemann ratio [see, for example, PRB 41, 796(R)]. Do the authors think that their theory could be extended to study these systems?

2. What are the prospects for using this approach in conjunction with first-principles methods? Could this become a predictive theory, for example for the study of high pressure systems? It is briefly mentioned in the manuscript that the existence of isomorphs has been established in some metallic systems using first-principles techniques, but expanding upon this point could be of interest to the large community of first-principles simulations.

REVIEWERS' COMMENTS:

Reviewer #1 (Remarks to the Author):

My comments and suggestions have been considered and taken into account by the authors in the revised version of the work. All points have been satisfactorily addressed. I recommend acceptance of the paper for publication in Nature Communications.

Reviewer #2 (Remarks to the Author):

I do not have much to add to my first report.

The paper is very nice and elegant, and the authors made a good job at taking into account all referee's comments.

However, I remain of my initial opinion that the paper is a (very nice and elegant) exercise in the context of a theory that has been well developed in the past years. As such, I do not see here the originality that is required by Nat.Comm.

Also, I understand the claim of generality made by the authors, but it should be kept in mind that many interesting potentials do not obey approximate scale invariance, and therefore the theory presented here is still limited to a quite special class of potentials.

All this said, my advice seem to be in the minority, so I guess that the paper will be accepted and I have no particular objection to this.

Reviewer #3 (Remarks to the Author):

The authors of the manuscript "Thermodynamics of freezing and melting" have satisfactorily addressed all points raised by the referees. I think this is an important contribution to the theory of freezing and melting that will motivate significant amounts of work in the area.

Resubmission of “Thermodynamics of freezing and melting” by Pedersen *et al.*

We wish to thank all three reviewers for their most useful critiques, which have led to a substantial improvement of the paper. In addition, we have taken the opportunity to polish the presentation here and there, and also added some new references.

Please find below our answers/comments and a list of the changes made accordingly. The reviewer comments are reproduced in *Italic*.

Reviewer #1 (Remarks to the Author):

The work deals with one of a most common and fundamental phenomenon encountered in the nature - the freezing/melting transition. This type of the phase transition has been a subject of intensive studies of many researches for decades and still remains unsolved problem.

The work develops the framework which provides the basis for calculating the fundamental thermodynamic properties of freezing and melting. The proposed method applies to the considerable class of systems and thus, contributes significantly to the basic research on the melting/freezing issue. It is based on the concept of the hidden scale-invariance and predicts the pressure, density and entropy along the melting line as well as the Lindemann ratio, viscosity and diffusion constant.

In my opinion the work can be of interest not only to the physicists and deserves to be published in Nature Communications.

We thank the reviewer for these positive words; in particular because, even though it is of fundamental importance, research into the thermodynamic fundamentals of freezing and melting is not fashionable at the moment (atomistic simulations certainly are).

The only my concern is the Supplementary Information which constitutes an essential part of the main text and in fact it is difficult to treat it as a supplementary material.

Some trivial algebra in the SI should be omitted (e.g., all the intermediate steps in getting eqs. 6,9,13.).

We agree that a thorough understanding of the theory is not possible without reading the Supplementary Information. All details could not be given in the main paper. In view of this we decided to give the details in the Supplementary Information and merely present a sketch of the line of reasoning in the main paper. There may, however, have been too many trivial details in the supplement, and we have followed the recommendation of omitting the algebra in getting Eqs. (6), (9), and (13).

There are several compact phrases and jargon words which need to be changed or explained, some examples are listed below.

p.2 "the canonical Lennard-Jones" - is too shortened and thus unclear

We have deleted the word “canonical” which may confuse the reader. The Lennard-Jones pair potential itself is defined in the next line.

p.2 "longer ranged attractive London(van der Waals) forces" - London forces are part of the van der Waals forces

That is correct, and we have now deleted the reference to London forces.

p.3 lambda in eq.(1) is to be explained

We now explain the interpretation of lambda below Eq. (1).

p.4 in Fig.1 caption as well as in the main text, "Structure" may need some extra comment what it denotes in this work.

We agree that this was inaccurate and have added the following sentence at the end of the paper's second paragraph:

“The term “structure” generally refers to the entire collection of spatial equal-time density correlation functions, but our focus below is on the pair correlation function as the most important structural characteristic(in the form of its Fourier transform, the structure factor).”

Also, in the caption for Fig. 1 the word “structure” has been qualified by adding “as reflected in the structure factor”. Hopefully, these changes clarify how the term structure is being used.

p.4 the last sentence "The fact lies ... the melting is unique." is not obvious what the authors mean here.

We have reformulated the sentence and now write (end of first paragraph, p. 5)

“The theory presented below makes use of the simplicity provided by R simple systems' almost one-dimensional phase diagram [38] and gives a correction to the hard-sphere picture of melting and freezing based on first-order Taylor expansions.”

p.2 in SI, "From an NVT equilibrium simulation" - a method which maintains constant T in calculations should be given.

We have added that the simulations were performed using a Nose-Hoover thermostat.

p.2 in SI, the average $\langle \dots \rangle_i$ in eq.(1) is not defined; also, one can argue that the standard expression is the ratio of averages not the average of the ratios,

We now define the average and have changed the equation such that it is now a ratio of two averages (as in the Landau-Lifshitz derivation of the configurational temperature, Eq. (33.14) in their book on statistical physics).

p.3 in SI, do the identity for the virial W implies U in eq.(4)

We are not quite sure about this question; the quoted identity for W implies the expression for W given in Eq. (4) (second line).

p.6 "In the general case" - it is not obvious what "general case" is here mentioned

We have added the clarifying remark:

“in which there is no analytical expression for the potential energy as a function of density,”

p.7 in SI, "A coefficients determined via eq.(9)" - in eq.(9) A' are determine; note in the text many times A instead of A' or A'' are called

This error has now been rectified.

Reviewer #2 (Remarks to the Author):

A. Summary of key results

In this paper, the authors present a theory of thermodynamics and dynamics along the melting line of a simple model of monoatomic liquid, the Lennard-Jones system.

The theory allows one to compute many observables of interest, including the diffusion constant, the viscosity, the Lindemann ratio, as well as the transition line itself in the (P,T) and (ρ,T) phase diagram.

Our theory is general and not just limited to the Lennard-Jones system.

B. Originality and interest

The paper is strongly based on previous results of the same authors about the concept of "hidden scale invariance" (e.g. refs. 38,39). I do not really see a major conceptual advance with respect to previous work.

It is correct that the paper is based on the previously developed isomorph theory. By now this theory has been applied to atomic and molecular systems, explaining their density-scaling thermal-equilibrium properties, and it has also been applied to non-equilibrium phenomena like the shear flow of liquids and glasses, to crystals, to nano-confined systems, to polymers, etc.

What sets the present work apart from all these developments is the following. Previously we always discussed melting and freezing from the assumption that the freezing/melting lines are exact isomorphs. Inspired by simulations of Heyes and coworkers, by own simulations, and by in-house experiments by Adrjanowicz and Niss, we have now reconsidered the problem. The encouraging result is that, while neither the freezing nor the melting line are isomorphs, they are so close to being isomorphs that a simple first-order Taylor expansion can account for most deviations from the many previously known freezing/melting line “invariants”. For instance, Luo *et al.* [J. Chem. Phys. **122**, 104709 (2005)] found that the Lindemann ratio is pressure dependent; we can now explain that quantitatively – among many other things.

The major conceptual advance of the paper, in our opinion, is that this is the first time the isomorph theory has been used to go beyond the theory itself. Several of our colleagues have throughout the years asked whether some sort of perturbation expansion may be developed in which the isomorph theory is the leading term, but only now do we have results showing that this is possible. This opens for novel uses of the isomorph theory.

Another major conceptual advance of the paper, in our opinion, is that people have never before attempted to, let alone succeeded in, predict freezing and melting line characteristics from properties of a single coexistence state point.

A final conceptual advance – not discussed in the paper but nevertheless potentially important – is that we can now predict the thermodynamics for a large part of the phase diagram from simulations carried out at a single state point. This goes much beyond freezing and melting.

C. Data & methodology

The theory seems robust, and the results are sound and accurate.

The presentation is clear, but in some points it is not completely clear what is a "rigorous" result and what is an assumption. For example, Eq.1 is stated to be mathematically equivalent to the same relation but with an equality. Is this really true? For what class of functions $U(\mathbf{R})$? And if it is true, wouldn't it be better to use directly the relation with equality in Eq.1, which would provide a rigorous justification for the scale invariance of dynamics and thermodynamics?

The mathematical equivalence between Eq. (1) and the formulation in the text with an equality sign instead of the inequality sign is exact for all functions $U(\mathbf{R})$. This follows from the fact that if Eq. (1) applies for all configurations and if two configurations have the same potential energy, their

scaled versions cannot have different potential energies (if they had, by scaling “backwards” one would conclude by Eq. (1) that the original configurations also has different potential energy, a contradiction).

The reason we used Eq. (1) and prefer that definition is that when it comes to systems that only obey Eq. (1) approximately, the inequality scaling condition of Eq. (1) may still apply to a good approximation, whereas the equality condition will break down at any minor deviations. Thus the property of approximately obeying Eq. (1) is not equivalent to approximately obeying the equality scaling condition. – This discussion deserves to be given in more detail than has hitherto been done, we agree in that, but the present paper does not seem to be the right place to do this.

Another minor remark is that on page 2 the authors write "the Lindemann ratio is predicted to be universal...". But this is a trivial consequence of the Hard Sphere model, right? Since HS have a single melting point, the Lindemann ratio can only take a single value.

This is correct – the HS model has a unique Lindemann melting ratio. We have modified the explanatory text of point 1 on p. 2 to state this more clearly by writing (top of p. 3)

“In the hard-sphere model the Lindemann ratio is universal at melting because there is just a single melting point. In particular, the HS model implies the Lindemann ratio to be invariant along the melting line for any given system.”

D. Statistics and uncertainties

This has been discussed correctly.

E. Conclusions

In conclusion, this is a nice and elegant work where the authors derive many properties of the Lennard-Jones system at melting from a set of simple assumptions and few numerical simulations. At the same time, the work looks like an "exercise" in the context of the general hidden scale invariance approach previously developed by the authors, rather than a major conceptual advance. For this reason, I am not sure that the paper meets the criteria of originality that are required by Nature Communication. The paper might be more appropriate for a specialised journal like J.Chem.Phys.

As pointed out by reviewers 1 and 3, the theory developed holds for many systems. Reviewer 3 also mentioned that the theory holds promise of significantly reducing the computational cost of studying melting.

The major message of the paper is that there is now, for the first time, a quantitatively accurate theory of the thermodynamics of melting. We feel that this in itself may be regarded as a major conceptual advance. Moreover, the fact that knowledge of a single coexistence state point is enough to determine everything (for the Lennard-Jones system) is novel.

The theory was derived by making use of the isomorph theory used already in several previous publications of ours, but the present paper is the first time that it has been possible to go beyond isomorph theory predictions.

Our previous work could be (very roughly) summarized as: 1) demonstrating the existence of isomorphs in various systems, 2) developing analytical forms for their shape, and 3) quantifying the degree to which formally isomorph invariant properties are in fact invariant along isomorphs. The analytical theory presented here allows us, at least for generalized LJ systems, to do something quite different: to predict the variation of quantities such as pressure, free energy, bulk modulus and more, along isomorphs. Previously all we could say was that such quantities are not isomorph invariant. In this work we apply this formalism to melting line properties, but it can in fact be used to predict all thermodynamic quantities in the whole part of the phase diagram where there are isomorphs, based on simulations along one isotherm, and also predict where the region of good isomorphs ends. This will be documented in future publications and has not been discussed in the paper (in order not to interrupt the flow).

Reviewer #3 (Remarks to the Author):

The manuscript describes a new methodology for the theoretical and computational study of freezing and melting. The authors demonstrate that, for a large class of systems, it is possible to use the properties of a single state point to calculate a large range of properties related to melting at other state points. These properties include, for example, the entropy of fusion and the Lindemann ratio.

More concretely, the starting point of the present work are a class of systems exhibiting "hidden scale-invariance". These systems have been extensively studied in recent years, with many contributions from the authors of the present paper. The novel contribution of the present manuscript is to exploit the properties of this class of systems to propose a new theory that, from the properties at a particular state point, those at other state points can be calculated. This approach holds promise for significantly reducing the computational cost of the study of melting.

We are actually working precisely in this direction. As we see it, the present paper opens an entirely new research direction in the modeling of freezing and melting.

The theory is adequately presented, with some of the more technical details left for the supplementary, which allows for a good flow of the main manuscript. The validation and usefulness of the new theory is demonstrated using the Lenard-Jones system as a test case. The results are convincing, and should motivate further work to expand them to other, more realistic systems.

Overall, the manuscript is well-structured, the main ideas are presented in a coherent and clear manner, and the conclusions are strong. Nonetheless, I would like to make a few suggestions to the authors:

1. Regarding the realm of applicability of the theory, could the authors include a discussion on quantum melting? Some of the references provided by the authors are for high pressure research. In this field, one of the outstanding questions is the possible existence of a zero-temperature quantum fluid of metallic hydrogen. The quantum nature of the proton modifies classical melting, for example by increasing the value of the Lindemann ratio [see, for example, PRB 41, 796(R)]. Do the authors think that their theory could be extended to study these systems?

This is a very interesting question; in fact one of our present lines of research deal precisely with to which degree isomorph invariance can have consequences for quantum liquids or solids. For instance, does the intriguing quasiuniversality of quantum liquids close to a Feshbach resonance discussed by T.-L. Ho, PRL **92**, 090402 (2004) have a connection to isomorph theory? Which simplifications appear for quantum versions of classical systems with hidden scale invariance?

For lack of space the present paper cannot contribute much to this discussion, but we now mention the question of a possible zero-temperature quantum fluid of metallic hydrogen by the following remark in the paper's final paragraph:

“It is not clear to which degree the theory may be generalized to quantum systems for which an outstanding question is the possible existence of a zero-temperature quantum fluid of metallic hydrogen. The quantum nature of the proton modifies classical melting, for example by increasing the value of the Lindemann ratio \cite{chu90}. It would be interesting to investigate whether melting of quantum crystals may be understood in the above framework, but at the moment this awaits the development of an isomorph theory for quantum systems.”

2. What are the prospects for using this approach in conjunction with first-principles methods? Could this become a predictive theory, for example for the study of high pressure systems? It is briefly mentioned in the manuscript that the existence of isomorphs has been established in some metallic systems using first-principles techniques, but expanding upon this point could be of interest

to the large community of first-principles simulations.

The reviewer here points in the direction of our future research plans: After recently having studied 58 elements by *ab initio* quantum DFT simulations in a collaboration with Georg Kresse and Felix Hummel in Vienna [PRB **92**, 174116 (2015)], we now hope to generalize the present paper's approach to deal with such systems. The challenge, of course, is that the metals are not pair-potential systems and that, moreover, there is no analytical equation for the potential energy's density dependence. – The paper ends by the following:

“In ongoing work we are addressing another open question, namely whether the above can be generalized to deal with more realistic systems, in particular metals for which density functional theory (DFT) computer simulations nowadays give realistic representations of the physics and have demonstrated hidden scale invariance \cite{hum15}.”